# Enhancement of Rydberg-mediated single-photon nonlinearities by electrically tuned Förster resonances

H. Gorniaczyk[1], C. Tresp[1], P. Bienias[2], A. Paris-Mandoki[1], W. Li[3], I. Mirgorodskiy[1], H.P. Büchler[2], I. Lesanovsky[3] & S. Hofferberth[1]

Mapping the strong interaction between Rydberg atoms onto single photons via electromagnetically induced transparency enables manipulation of light at the single-photon level and few-photon devices such as all-optical switches and transistors operated by individual photons. Here we demonstrate experimentally that Stark-tuned Förster resonances can substantially increase this effective interaction between individual photons. This technique boosts the gain of a single-photon transistor to over 100, enhances the non-destructive detection of single Rydberg atoms to a fidelity beyond 0.8, and enables high-precision spectroscopy on Rydberg pair states. On top, we achieve a gain larger than 2 with gate photon read-out after the transistor operation. Theory models for Rydberg polariton propagation on Förster resonance and for the projection of the stored spin-wave yield excellent agreement to our data and successfully identify the main decoherence mechanism of the Rydberg transistor, paving the way towards photonic quantum gates.

[1] 5th Institute of Physics and Center for Integrated Quantum Science and Technology, Universität Stuttgart, Pfaffenwaldring 57, 70569 Stuttgart, Germany. [2] Institute for Theoretical Physics III and Center for Integrated Quantum Science and Technology, Universität Stuttgart, Pfaffenwaldring 57, 70569 Stuttgart, Germany. [3] School of Physics and Astronomy, University of Nottingham, Nottingham NG7 2RD, UK. Correspondence and requests for materials should be addressed to H.G. (email: h.gorniaczyk@physik.uni-stuttgart.de) or to S.H. (email: s.hofferberth@physik.uni-stuttgart.de).

Rydberg excitations of ultracold atoms[1] are currently attracting tremendous attention because of possible applications in quantum computing[2–5] and simulation[6–10]. One particular aspect is the realization of few-photon nonlinearities mediated by Rydberg interaction[11–14], enabling novel schemes for highly efficient single-photon generation[15,16], entanglement creation between light and atomic excitations[17], single-photon all-optical switches[18] and transistors[19,20], single-photon absorbers[21] and interaction-induced photon phase shifts[22,23]. Interacting Rydberg polaritons also enable attractive forces between single photons[24], crystallization of photons[25] and photonic scattering resonances[26]. The above experiments and proposals make use of the long-range electric dipole–dipole interaction between Rydberg atoms[27–31]. A highly useful tool for controlling the interaction are Stark-tuned Förster resonances, where two dipole-coupled pair states are shifted into resonance by a dc[32] or microwave[33,34] electric field. Förster resonances have been studied by observation of dipole blockade[35], line shape analysis[36], double-resonance spectroscopy[37], excitation statistics[38] and Ramsey spectroscopy[39,40]. Recently, resonant four-body interaction[41] and the anisotropic blockade on Förster resonance[42] and quasi-forbidden Förster resonances[43] have been observed, and Förster resonances between different atomic species have been predicted[44]. For Rydberg-mediated single-photon transistors, the near-resonance in zero field for specific pair states has been used to enhance the transistor gain[20], while in experiments on Rydberg atom imaging[45,46] an increase in Rydberg excitation hopping has been observed on resonance[47].

In this work, we use Stark-tuned Förster resonances to greatly increase the interaction between individual photons inside a Rydberg medium. We achieve this by tuning pair states $|S^{(g)},S^{(s)}\rangle$ containing two different Rydberg S-states into resonance with $|P^{(g)},P^{(s)}\rangle$ pair states by an electric field. We show that for gate and source Rydberg states $|50S_{1/2},48S_{1/2}\rangle$, we can boost the performance of a Rydberg single-photon transistor. When operated classically, we achieve $\mathcal{G} > 100$, enabling high-fidelity detection of single Rydberg atoms. This improved transistor can be operated such that the gate photon is read out with finite efficiency, reaching a gain $\mathcal{G} > 2$. We develop theoretical models for the dynamics of Rydberg polaritons in the presence of Förster resonances and the loss of coherence due to photon scattering. Excellent agreement with our experimental data is found. Finally, our all-optical probe represents a novel approach for the high-resolution study of the substructure of Förster resonances caused by fine structure and Stark/Zeeman splitting of the $|P^{(g)},P^{(s)}\rangle$ pair states. We demonstrate this technique by resolving the multi-resonance structure of the $|66S_{1/2},64S_{1/2}\rangle$ pair state.

## Results

**Experimental set-up.** Our experimental scheme[13,19,20,45] is shown in Fig. 1a,b: by coupling the excited state $|e\rangle$ and the Rydberg state $|S^{(g)}\rangle$ with a strong light field $\Omega_g$ with detuning $\delta_g$, a gate photon $\mathcal{E}_g$ is converted into a Rydberg excitation inside a cloud of ultracold $^{87}$Rb atoms. We then probe the presence of this gate excitation by monitoring the transmission of source photons $\mathcal{E}$ coupled via electromagnetically induced transparency (EIT) to the source Rydberg state $S^{(s)}$. Specifically, we use ($\delta_g = 40$ MHz) for efficient Raman absorption of the gate photon in the experiments without retrieval, while we use EIT-based slow light techniques ($\delta_g = 0$) for photon storage in experiments with gate photon retrieval. At zero electric field, the interaction between the $|S^{(g)},S^{(s)}\rangle$ pair is of van der Waals type. The difference in electric polarizability between S- and P-states enables the shift of the initial pair state into degeneracy with specific $|P^{(g)},P^{(s)}\rangle$ pairs, resulting in resonant dipole–dipole

interaction. We shift the Rydberg levels by applying a homogeneous electric field along the direction of beam propagation. Active cancellation of stray electric fields is done with eight electric field plates in Löw configuration[48], while the homogeneous field results from additional voltages $V^+, V^-$ to four electrodes (Fig. 1a).

**Stark-tuned optical nonlinearities.** We first study the pair state $|S^{(g)},S^{(s)}\rangle = |66S_{1/2},64S_{1/2}\rangle$. Due to the fine structure splitting of the Rydberg P-states, this pair is near resonant with two P-state pairs $|65P_{1/2},64P_{3/2}\rangle$ and $|65P_{3/2},64P_{1/2}\rangle^{20}$. Both $|P^{(g)},P^{(s)}\rangle$ pairs can be tuned into resonance at electric fields $\epsilon < 0.25$ V cm$^{-1}$. The full pair state Stark map in the presence of a magnetic field $B = 1$ G (Fig. 1c, gray lines) reveals a large number of closely spaced resonances arising from the non-degenerate $(m_j^{(g)}, m_j^{(s)})$ combinations. The strength of individual resonances depends on the angle $\theta$ between the interatomic axis and the quantization axis defined by the external fields, resulting in a non-spherical blockade volume[29]. We explore these resonances by measuring the optical gain

$$\mathcal{G} = \left( \bar{N}_{s,out}^{no\ gate} - \bar{N}_{s,out}^{with\ gate} \right) / \bar{N}_{g,in}, \qquad (1)$$

that is, the mean number of source photons scattered by a single incident gate photon[20], as a function of applied electric field (Fig. 1c). Our high-resolution spectroscopy indeed reveals four resonances, matching with the calculated crossings of different pair state groups. In between resonances, the coupling of $|S^{(g)},S^{(s)}\rangle$ to multiple $|P^{(g)},P^{(s)}\rangle$ pair states with positive and negative Förster defects results in smaller blockade than in the zero-field case. This interplay between different resonances actually decreases the measured gain with respect to the field-free value. This situation does not occur for the Förster resonance $|50S_{1/2},48S_{1/2}\rangle \leftrightarrow |49P_{1/2},48P_{1/2}\rangle$ at $\epsilon < 0.710$ V cm$^{-1}$ (Fig. 1d). For this state combination there is one isolated resonance, resulting in the single peak in the optical gain.

**Rydberg polaritons near Förster resonance.** To quantitatively describe the observed resonances, we include in the microscopic description of polariton propagation[13,14,26] the special character of the interaction close to Förster resonance, see Supplementary Note 1. For illustration, we consider the $|50S_{1/2},48S_{1/2}\rangle$ pair and angle $\theta = 0$, which results in the selection rule $\Delta M = \Delta m_j^{(g)} + \Delta m_j^{(s)} = 0$ for the magnetic quantum numbers of the involved states. We then need to include four pair states: $\{|50S_{1/2},48S_{1/2}\rangle,\ |49P_{1/2},48P_{1/2}\rangle,\ |48P_{1/2},49P_{1/2}\rangle,\ |48S_{1/2},50S_{1/2}\rangle\}$ with $(m_j^{(g)}, m_j^{(s)}) = (\frac{1}{2}, \frac{1}{2})$. In this basis, the interaction Hamiltonian reduces to

$$H_{dd}(r) = \frac{1}{r^3} \begin{pmatrix} 0 & C_3 & C_3' & 0 \\ C_3 & 0 & 0 & C_3' \\ C_3' & 0 & 0 & C_3 \\ 0 & C_3' & C_3 & 0 \end{pmatrix} \qquad (2)$$

with two dipolar coupling parameters $C_3$, $C_3'$. Since the interaction is dominated by the Förster resonance, we neglect any residual van der Waals interactions. In general, the Hamiltonian (2) gives rise to flip-flop (hopping) processes of type $|50S_{1/2},48S_{1/2}\rangle \rightarrow \{|49P_{1/2},48P_{1/2}\rangle,\ |48P_{1/2},49P_{1/2}\rangle\} \rightarrow |48S_{1/2},50S_{1/2}\rangle$. However, for this choice of Rydberg states the dipolar coupling parameters satisfy $C_3 \gg C_3'$, and therefore provide a strong suppression of hopping[49]. This behaviour is in contrast to the results in ref. 47, where hopping processes strongly influenced the interaction-mediated imaging of Rydberg excitations. In the experimentally relevant regime with $\omega, \gamma_s, \gamma_p \ll \Omega, \gamma$, where $\omega$ is the source photon detuning, while $\gamma_s$ and $\gamma_p$ describe the decoherence

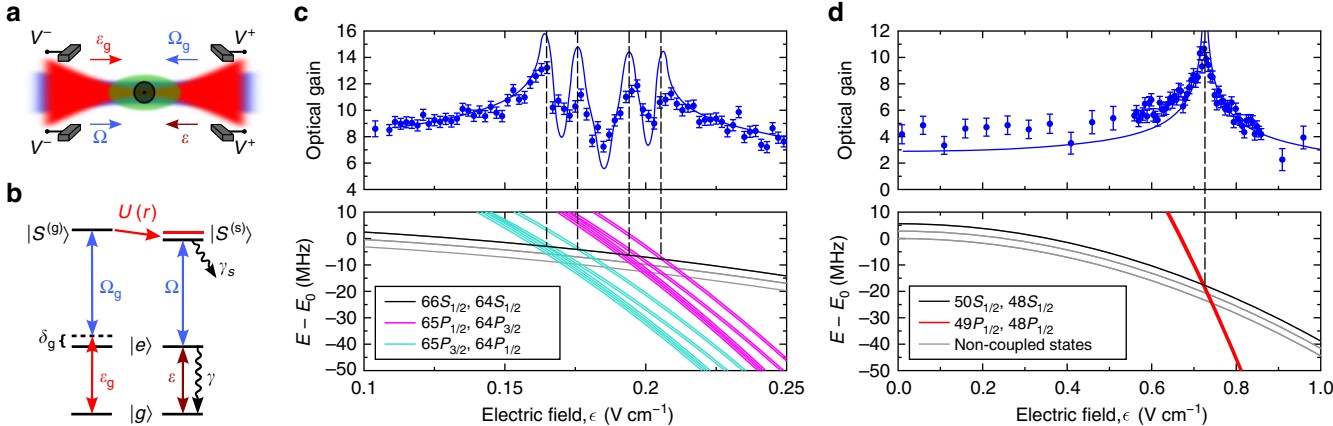

**Figure 1 | High-resolution spectroscopy of Förster resonances. (a)** Tightly focussed source and gate beams ($w_0 = 6.2\,\mu m$) are overlapped with an optically trapped cloud of $2 \times 10^4$ $^{87}$Rb atoms at $3\,\mu K$ (cylindrical $1/e$ dimensions $L = 40\,\mu m$, $R = 10\,\mu m$). For each transistor operation the optical trap is shut off for $200\,\mu s$. We perform 23 individual experiments in a single cloud, recapturing the atoms in-between with minimal loss and heating. In-vacuum electrodes are used to apply the electric field. **(b)** Level scheme for gate and source photons coupled to different Rydberg states, where $2\Omega$ is the Rabi frequency of the control field and $2\gamma$ is the decay rate of $|e\rangle$. **(c,d)** At certain electric fields (vertical dashed lines), the $|S^{(g)}, S^{(s)}\rangle$ pair state is resonant to pair states of type $|P^{(g)}, P^{(s)}\rangle$. The enhancement of interaction between $|S^{(g)}\rangle$ and $|S^{(s)}\rangle$ manifests in peaking of the transistor gain (blue dots). In **c**, the fine structure of the involved $P$-states and the $m_J$-dependence of the Stark-shift result in the observed multi-resonance structure. The blue solid line is a theoretical analysis of the full-polariton propagation in the presence of the gate excitation. The error bars are the s.e.m.

rates of $|S^{(s)}\rangle$ and $|P^{(s)}\rangle$ excitations, the equation describing a single polariton $\mathcal{E}(r, \omega)$ and its interaction with the gate Rydberg excitation $|S^{(g)}\rangle$ at position $r_j$ simplifies to

$$\left( ic\partial_r + \frac{g^2(\omega - i\gamma_s)}{\Omega^2} + \frac{g^2 V_{ef}^j(r)}{\Omega^2 - i\gamma V_{ef}^j(r)} \right) \mathcal{E}(r, \omega) = 0 \quad (3)$$

as derived in our Supplementary Note 1. Here $g = g_0 \sqrt{n_{at}}$ is the collective coupling strength with $g_0$ being the single-atom–photon coupling strength and $n_{at}$ is the atomic density. The effective interaction $V_{ef}^j$ simplifies to

$$V_{ef}^j(r) = \frac{C_3^2}{\Delta_D - \omega - i\gamma_p} \frac{1}{(r - r_j)^6} \quad (4)$$

where $\Delta_D$ is the Förster defect. It is remarkable that, regardless of $\Delta_D$, our microscopic derivation provides an effective interaction always based on van der Waals type interaction.

For comparison with experiment, we generalize our calculation to nonzero angles $\theta$ between the quantization and interatomic axis, as well as to the larger number of states involved for the $|66S_{1/2}, 64S_{1/2}\rangle$ pair. We then integrate equation (3) over the cloud shape and average over the stored spin-wave. We also take into account the Poissonian statistics of the gate and source photons, the storage efficiency, the fact that the blockade radius is comparable to the beam waist and the finite experimental resolution in electric field $\Delta\epsilon = \pm 2\,mV\,cm^{-1}$, see Supplementary Note 1. The comparison, without any free parameters, with experimental results for the gain is shown in Fig. 1. We find very good agreement for all electric fields except very close to the resonances. One reason for the discrepancy is the following: Close to the Förster resonance and for distances on the order of $r_b$ between gate and source, the atomic part of the polariton-excitation pair initially in $|50S_{1/2}, 48S_{1/2}\rangle$ is converted into the superposition of $|49P_{1/2}, 48P_{1/2}\rangle$ and $|50S_{1/2}, 48S_{1/2}\rangle$. This results in additional slowing down of the polariton, and, consequently, an accumulation of polaritons close to $r_b$. Then, the assumption to study the propagation of individual polaritons breaks down as the interaction between the polaritons has to be included.

**Resonant single-photon transistor.** Next, we investigate to what extent these Förster resonances can be used to improve the Rydberg single-photon transistor[19,20]. We find that for this application, the $|50S_{1/2}, 48S_{1/2}\rangle$ resonance is ideal. It enables large-source photon input rates, because of the relatively weak van der Waals interaction between two source photons. On the other hand, the Förster resonance provides sufficient gate–source interaction to observe high transistor gain. For source photon rate $R_{in} = 35\,\mu s^{-1}$, we reach a maximal gain of $\mathcal{G} = 200$. At such high source rates, we observe small temporal changes in transmission, which we attribute to an accumulation of stationary Rydberg excitations in the medium caused by dephasing of single-source polaritons. This effect has been previously observed for Rydberg $S$-states[14] and differs from the interaction-induced dephasing of $D$-state polariton pairs[50]. This accumulation sets an upper limit on the source photon rate for the non-destructive imaging of single Rydberg excitations[45], since the creation of additional Rydberg atoms also 'destroys' the original system. We thus restrict our analysis in Fig. 2 to non-destructive source input rates for which the maximum temporal change in source transmission remains $< 10\%$. In this regime, we observe a linear increase of the optical gain with $R_{in}$ both at zero electric field and on the Förster resonance (Fig. 2a). Exploiting the Förster resonance, we can improve the optical gain by a factor $> 2$ on resonance (blue dots) compared with the zero-field case (blue squares). The large number of source photons scattered from a single-gate excitation enables the single-shot detection of a stored gate photon with high fidelity[18,19,51], see Methods. In Fig. 2, we show this fidelity as a function of the applied electric field for two source photon rates. The Förster resonance enables a substantial increase of the fidelity to a maximal value of $\mathcal{F} = 0.8$. This number is mainly limited by the fact that our beam waist $w_0$ is slightly larger than the gate–source blockade distance. For spatially resolved Rydberg detection[45,46], even higher fidelities are possible using imaging systems with better optical resolution than our beam size $w_0 = 6.2\,\mu m$.

**Single-photon transistor with gate photon read-out.** The improved gate–source interaction on Förster resonance enables us for the first time to operate our transistor with retrieval of the

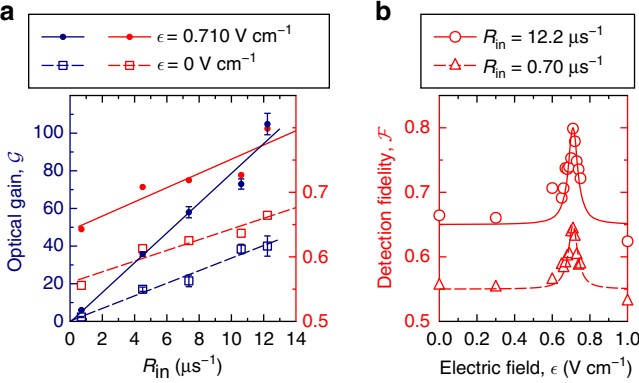

**Figure 2 | Transistor gain and single Rydberg detection.** Performance of the single-photon transistor on the $|50S_{1/2},48S_{1/2}\rangle \leftrightarrow |49P_{1/2},48P_{1/2}\rangle$ resonance. (**a**) Gain and single Rydberg detection fidelity increase linearly with the rate of incident source photons $R_{in}$ in the non-destructive range, where the creation of stationary excitations from source photons is negligible. Both the optical gain (**a**) and the single Rydberg detection fidelity (**a,b**) are highly amplified on the Förster resonance at $\epsilon = 0.710$ Vcm$^{-1}$. The solid curves are linear or Lorentzian fits to guide the eye. The error bars are the s.e.m.

stored gate photon after the transistor operation[51]. To store the gate photon, we stop the polariton inside the medium by ramping down the control field $\Omega_g$ to zero for $\delta_g = 0$. Conversely, to read out the gate photon, $\Omega_g$ is turned on again. Without any source photon input between the storage and the read-out, we measure a lifetime of 3.6 μs for the atomic coherence of the stored gate spin-wave, mainly limited by the finite temperature of our atomic sample. Next, we apply a source pulse containing a mean number of photons $\bar{N}_{in}$ and pulse length $T = 3.2$ μs during a storage time of 4.2 μs. On Förster resonance, we achieve a mean number of scattered source photons within this time of up to 2.7 photons for a single stored gate photon (Fig. 3a). This is the first demonstration of a transistor with gain $\mathcal{G} > 2$ and read-out, a fundamental step towards quantum circuits employing feedback and gain or the non-destructive detection of the gate photon[52].

The overall fidelity of the transistor is limited by projection and dephasing of the gate spin-wave due to scattered and transmitted source photons[51,53]. In Fig. 3a, we show the absolute retrieval efficiency versus incident and scattered source photons at a mean number of $\bar{N}_{g,in} = 0.8$ incident gate photons on and off the Förster resonance. Interestingly, both cases collapse onto one exponential decay if plotted versus the number of scattered source photons. The black curve in Fig. 3a assumes zero retrieval fidelity for one or more scattered source photons. The dotted line and the dashed line, on the other hand, investigate the other hypothetic cases that the coherence of the gate spin-wave is destroyed by one photon of $\bar{N}_{in}$ incident mean photons (dashed) and by one photon of $\bar{N}_{in}^{r_b}$ mean photons incident on the blockade sphere (dotted), respectively. By applying established theory to our data in the next section, we will show that both transmitted photons and scattered photons contribute to the coherence and thus to the retrieval efficiency of the stored spin-wave.

**Theory on coherent spin-waves.** For more quantitative analysis we follow ref. 53, considering a one-dimensional (1D) model of the zero-field case for a single-source photon passing through the atomic cloud with Gaussian density profile. The gate photon is stored in the initial spin-wave state $\hat{\rho}_i$ and interacts with source photons via the potential from equation (4). After the source photon has left the atomic cloud, the state of the atomic ensemble is $\hat{\rho}_f$, and the quantum mechanical fidelity between the initial and final state is given by $F = \left[\text{Tr}\left|\sqrt{\hat{\rho}_i}\sqrt{\hat{\rho}_f}\right|\right]^2 = F_p + F_s$ (ref. 54).

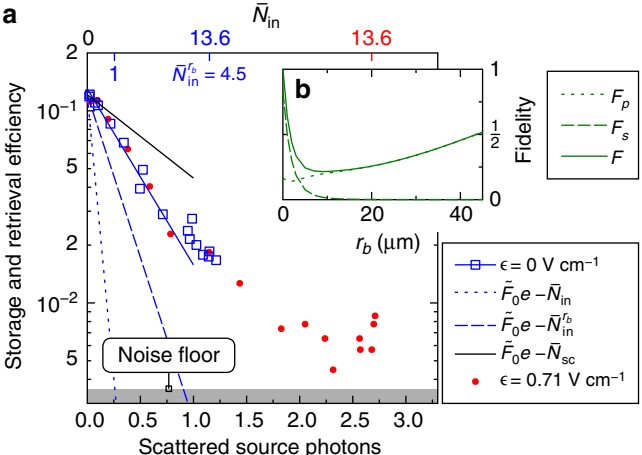

**Figure 3 | Transistor operation with retrieval of the gate photon.** (**a**) Efficiency of storing and reading out one single gate photon versus the number of scattered source photons during the storage time of 4.2 μs. When plotted as function of scattered photons, the observed retrieval efficiencies on Förster resonance (red dots) and in zero field (blue squares) are identical. (**b**) Calculated fidelity, that is, the overlap between the initial gate spin-wave state and the final state after the propagation of a source photon through a one-dimensional Gaussian atomic cloud. The fidelity is the sum of contributions from scattered (short dashes) and transmitted (long dashes) source polaritons. The lines in **a** show the predicted decay of retrieval efficiency using the full propagation model (solid blue line), as well as different limiting cases (see main text for details).

Here, $F_p$ accounts for transmitted and $F_s$ for scattered source polaritons. Both contributions are shown in Fig. 3b as a function of the blockade radius $r_b = (\gamma C_6/\Omega^2)^{1/6}$ for our experimental parameters. For large blockade radii, $F_p$ becomes negligible because source photons are rarely transmitted through the blockaded region. To describe the experimental 3D situation, we average the fidelities from Fig. 3b over the spatial transversal distribution of gate and source photons. With this approach, we obtain the blue solid line in Fig. 3a, which is in very good agreement with our data, despite the simplifications of our model. We consider this as evidence for the assumed mechanisms for the spin-wave decoherence to be correct. By identifying the decoherence mechanisms, we can isolate the required improvements for a high-fidelity coherent Rydberg transistor: the blockade volume of a single-gate excitation must be larger than the stored gate spin-wave to avoid the projection, while the optical depth $OD_B$ inside the blockaded region must be large to prevent the dephasing due to transmitted photons. Meeting both requirements simultaneously is challenging due to limits on the atomic density because of Rydberg-ground state interaction[18,55].

## Discussion

Rydberg-mediated single-photon nonlinearities can be greatly enhanced by electrically tuning adjacent pair states to Förster resonance. By carefully choosing the employed Förster resonance, we have simultaneously improved the Rydberg transistor gain and the fidelity of single Rydberg atom detection. We identify the $|50S_{1/2},48S_{1/2}\rangle \leftrightarrow |49P_{1/2},48P_{1/2}\rangle$ resonance in $^{87}$Rb as ideal both for the Rydberg single-photon transistor and non-destructive imaging of Rydberg atoms[45,46]. Exploiting this resonance, we have demonstrated the first operation of the Rydberg transistor with read-out of the gate photon. Our quantitative analysis of the reduction of retrieval efficiency caused by source photons points

the way towards high-fidelity Rydberg-based photonic gates and transistors. Our polariton propagation theory correctly accounts for the enhanced source–gate interaction and is in excellent agreement with the experiment. It also reveals unexpected and rich properties close to Förster resonances. This regime enables study of the transition from two- to many-body interaction and propagation with excitation hopping[47,56]. The complexity of the resonances due to the Rydberg-level structure provides a wide range of tuning options. The gate–source interaction can be reduced or even switched off between individual resonances. Similarly, by addressing different Zeeman pair state resonances with the external field the angular dependence of the interaction can be greatly varied. This provides a rich set of new tools for tailoring the interaction of photons coupled to different Rydberg states inside the medium.

## Methods

**Preparation of the ultracold atomic sample.** We load $^{87}$Rb from a constant Rubidium background pressure of $10^{-9}$ mbar atoms into a magneto-optical trap (MOT). Simultaneously, a crossed optical dipole trap (ODT) from a fibre laser at a wavelength of 1,070 nm superimposes the MOT and attracts atoms of both $5^2S_{1/2}$ hyperfine ground states ($F = 1,2$). After 1 s of loading, the MOT is compressed by ramping up the quadrupole magnetic field by a factor of 8 during 40 ms followed by an optical molasses phase of 5 ms, to maximize the number of atoms loaded into the ODT. The intensity of the ODT laser is ramped down within 200 ms to perform forced evaporation yielding $2 \times 10^4$ atoms with a temperature 3 µK in a cigar shaped trapping potential with a $1/e$ half length of $L = 40$ µm and a radius of $R = 10$ µm. Finally, by shining in two pumping lasers, we transfer atoms from the $F = 1$ state to the $F = 2$ state and optically pump the population to the stretched state $m_F = 2$.

**Probing the optical nonlinearity.** The gate excitation and the source EIT are realized with four independent laser systems, with the lower transition gate and source photons (near-)resonant to the MOT transition to achieve a maximum optical depth and thus highest efficiency of single-photon absorption. The upper transition is at 480 nm. All four laser beams are overlapped on one axis with polarization optics and dichroic mirrors. Achromatic lenses are used from both sides to focus and collimate the laser beams. The transmitted source and gate photons are coupled through single-mode fibres and detected on commercial avalanche photodiodes. Taking loss at optics and fibre coupling into account, photons in the experiment are detected with an efficiency of 30%.

**Data acquisition.** To reduce the statistical error, we average over multiple experiments. For instance, the data points in Fig. 1c are gathered during 23 transistor measurements per MOT cycle. We measure at 1 electric field during 20 MOT cycles. The same procedure is repeated for the reference measurement which contains no gate photons. The fields are scanned in a triangular electric field scan which was repeated 15 times. That way, systematic errors are suppressed. In addition, by monitoring the source transmission we make sure that electric field drifts are negligible during the measurement. A similar procedure was done to measure the data in Fig. 2, but with yet another scan dimension, the source photon rate.

**Single Rydberg detection.** The attenuation of many source photons due to one gate photon (gain) is used to predict the single-shot existence of a gate Rydberg excitation via the number of detected source photons. If a low number of source photons is detected, probably a gate excitation was present which attenuated the source. Likewise, if a high number of source photons is detected, probably the gate excitation was absent. To quantify the minimum probability of the correct prediction (detection fidelity), we take two histograms of detected source photons, with and without incident gate photons, respectively. With the knowledge of the storage efficiency (60%) and the Poissonian statistics of the coherent gate photons (mean value $\bar{N}_g = 1$), it is possible to separate the histogram with this mean gate photon input into two histograms, one corresponding to the events with no gate excitations present and one with gate excitations. With a discrimination line, we set a threshold value for the decision whether or not the excitation was present. Any overlap of both histograms through this line results in a fidelity $\mathcal{F} < 1$.

**Data availability.** The data that support the findings of this study are available from the corresponding author upon request.

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

## Acknowledgements

We thank Johannes Schmidt for construction of the electric field control; Sebastian Weber for calculation of Rydberg potentials; and Christian Zimmer for contribution to the experiment. This work is funded by the German Research Foundation through Emmy-Noether-grant HO 4787/1-1 and within the SFB/TRR21. H.G acknowledges support from the Carl-Zeiss Foundation. I.L. acknowledges funding from the European Research Council under the European Union's Seventh Framework Programme (FP/2007–2013)/ERC Grant Agreement No. 335266 (ESCQUMA), the EU-FET Grant No. 512862 (HAIRS), the H2020-FETPROACT-2014 Grant No. 640378 (RYSQ) and EPSRC Grant No. EP/M014266/1. W.L. is supported through the Nottingham Research Fellowship by the University of Nottingham and acknowledges access to the University of Nottingham HPC Facility.

## Author contributions

The experiment was conceived by H.G., C.T. and S.H. and carried out by H.G., C.T., A.P.-M., and I.M.; data analysis was done by H.G., A.P.-M. and C.T.; theory models and calculations were contributed by P.B., W.L., H.P.B., and I.L.; H.G. and S.H. wrote the manuscript with contributions from all authors.

## Additional information

**Competing financial interests:** The authors declare no competing financial interests.

