## [Peer review file · Nature Communications]

Reviewers' Comments:

Reviewer #1 (Remarks to the Author)

The authors present a study of applying electrically-tuned Förster resonances for single-photon optical transistor. When tuned to such resonances, they observe enhancement of transistor gain. This work follows prior related studies by the authors and by the Max Planck group [18-20], including application of a Förster resonance by the latter group [20]. The main difference in the current work is using electrical field to better tune to the Förster resonance.

The novelty and originality of the experimental approach compared to the prior works [18-20] are modest. The advance of applying electrical fields is mostly of technical nature and has been done previously, discussed in a review by Gallagher, T. F., and P. Pillet, 2008, "Dipole-Dipole interactions of Rydberg atoms," *Adv. At. Mol. Opt. Phys.* 56, 161.

This work does present improved transistor performance with respect to prior works although it is not clear that it is of qualitative nature rather than merely quantitative one.

In terms of presentation of the methods and results, I find the manuscript to be lacking detail and clarity. For example, when giving measured transistor gain, neither data for extinction nor the histogram of detected clicks are shown. These should have been at least included in the supplementary information. Likewise, the description of the experimental approach and sequence are rudimentary. These deficiencies complicate accessing the quality of the measured data and make it less useful for the reader.

Overall the manuscript contains new results and should be published. However in my opinion it does not clear the bar in terms of novelty and originality and quality of presentation for *Nature Communications*. I therefore do not recommend publication.

Reviewer #2 (Remarks to the Author)

This is a generally very nice presentation of an interesting experiment showing the enhancement of effective photon-photon interactions using Förster resonances.

There is one serious defect to the paper, which can be easily remedied. The paper simply cannot be understood without reading Refs 19-20. The paper heavily relies on a deep understanding of those papers.

Equation (3) is asserted without even a sketch of where it comes from, or a reference for a similar calculation elsewhere. The derivation should be either described in more detail, referenced to another work that will allow it to be understood, or included as Supplemental Material. The result that the effective interaction is van der Waals in its distance dependence even in the Förster regime is interesting and important.

With these two caveats, this is a beautiful piece of work and without question should be published in *Nature Communications*.

Reviewer #3 (Remarks to the Author)

The authors report in this manuscript on an improved version of their single-photon transistor, ref. [19]. In particular, electrically-controlled Förster resonances are used to increase the interaction strength between gate and source photons that are stored / propagating as collective Rydberg

polaritons within a cloud of Rubidium atoms. Due to the enhanced interaction, the transistor can be operated at a highly increased gain (>200), achieves high fidelities (>0.8) for detecting single Rydberg excitations, and makes it possible to investigate the coherence of the stored gate polariton after interaction with the source photons. The results are a significant improvement over the previous ones, and the work thoroughly investigates the limitations and decoherence effects of this system.

I think this work is a significant step forward towards applications of the single photon transistor as for instance in photonic circuits or for imaging single Rydberg excitations. As such, I am convinced that it will be very important and useful for other people working in this field. Thus, I believe it would be very well suited for publication in Nature Communications.

Nevertheless, I think the authors should formulate very carefully what they mean with coherence of their single-photon transistor. It seems to me that in their experiment the scattering of a single source photon completely destroys the coherence of the gate's collective Rydberg excitation (the data is below the black curve). The way I understand this data is that on average 2.7 photons are scattered, but sometimes much less than that. The remaining coherence is due to the fact that sometimes there is no photon scattered even for on average 2.7 photons. In my opinion the authors investigate the coherence of their single-photon transistor, but they do not show coherent operation. Nevertheless, I do understand the theory in such a way that a single scattered photon does not need to localize the position of the Rydberg excitation as long as the blockade radius is large enough (F_s increases for large r_b in Fig. 3b). Thus, coherent operation should be possible. Please note: A "coherent single-photon transistor" can also be understood as a transistor that operates coherently, i.e. a device where one uses a superposition of ground state and collective Rydberg excitation to control the transmission/reflection of the source photons. Thus, the GS would give transmitted photons, while the Rydberg excitation would result in scattered (reflected) photons, thus entangling the gate state with the path of the source photons. In any case I do not think that the expression "coherent single photon transistor" is suitable for this experiment.

Some further questions:

- How is the storage of the gate photon done? Is this based on EIT or just direct absorption?
- In Figure 2(a) I think it is not good to distinguish detection fidelity and optical gain only by colour (red/blue). For people that cannot distinguish these colours very well there is no way to see what is what.
- How is the detection fidelity measured? I suppose the method uses a threshold to distinguish the states similar to Fig. 4b in ref. [19]. In my opinion this should shortly be explained in the manuscript.
- How is the accumulation of stationary Rydberg excitations observed? I guess that the authors see a temporal change of the transmission during source irradiation and then assume that this is due to the accumulation of stationary Rydberg excitations. Please formulate more logical by describing observation and conclusion.

Reviewer #1:

The authors present a study of applying electrically-tuned Förster resonances for single-photon optical transistor. When tuned to such resonances, they observe enhancement of transistor gain. This work follows prior related studies by the authors and by the Max Planck group [18-20], including application of a Förster resonance by the latter group [20]. The main difference in the current work is using electrical field to better tune to the Förster resonance.

The novelty and originality of the experimental approach compared to the prior works [18-20] are modest. The advance of applying electrical fields is mostly of technical nature and has been done previously, discussed in a review by Gallagher, T. F., and P. Pillet, 2008, "Dipole-Dipole interactions of Rydberg atoms," *Adv. At. Mol. Opt. Phys.* 56, 161.

This work does present improved transistor performance with respect to prior works although it is not clear that it is of qualitative nature rather than merely quantitative one.

The previous experiments by the Max Planck group make use of a naturally occurring near-resonance in the absence of electric fields. For this purpose, the transistor gain is measured for different state combinations in the range $n \sim 65$. In our work, we electrically tune two Rydberg-states to exact resonance. The previous experiment shows that the interstate van-der-Waals interaction in zero field can have local maxima (on top of its overall scaling with increasing n).

In contrast, in our case of electrically tuned exact resonances, the interaction changes qualitatively from van-der-Waals to resonant dipole-dipole. This leads to a number of novel results we present in this paper:

- We indeed boost the demonstrated gain of the Rydberg single-photon transistor by more than a factor 10 compared to [19,20]. This point, on which the reviewer exclusively comments, is an important improvement for us and others working on optical transistors, but we agree that it in itself is more quantitative than a qualitative change.
- This improvement enables us to demonstrate for the first time an optical transistor with gain > 2 when the gate photon is retrieved, which is an important qualitative step forward towards transistor-based applications. In particular, with this improvement, we can thoroughly test the evolution of the coherence of the stored gate photon during the transistor operation. We successfully compare our data to recent theory, thus verifying experimentally the main decoherence mechanism. This investigation is of interest to the quantum information community since the study of decoherence mechanisms paves the way towards a quantum coherent optical transistor. More generally, the underlying decoherence mechanism is relevant for all optical gates based on Rydberg EIT, not just for the transistor.
- Our work is the first high-resolution spectroscopy of Förster resonances showing the effects of the fine-structure of the involved P-states. First, this establishes all-optical probing of these resonances as a new method. Secondly, this reveals the intricate interaction effects that can be tuned via electric fields, such as the reduction of the interaction exactly between two individual resonances. This has, to our knowledge, not been shown so far in any Rydberg experiment.
- Finally, we present a rather surprising theory result by analyzing Rydberg polariton propagation in the presence of a stored excitation in Förster-resonance with the polaritons. The obvious expectation, that the dipole-dipole interaction between stationary excitations is simply mapped onto the polariton-excitation interaction, turns out to be incorrect, as shown in the manuscript. Moreover, the standard assumptions in the polariton calculation break down close to the resonance, because the polaritons are extremely slowed and start to pile up close to the stored excitation. We consider this a rather surprising result, which we believe will stimulate further work on this topic.

We absolutely agree with the reviewer that electrically tuned Förster resonances are an established tool in Rydberg physics. This is why in our first submission we already dedicate a whole paragraph and the citations [27-41] to description of previous work (now we added J. H. Gurian et. al., *Phys. Rev. Lett.* **108**, 023005, 2012 to this list).

There are a number of thorough reviews on Rydberg interaction, we have added the reference mentioned by Reviewer #1 and (Comparat and Pillet, *J. Opt. Soc. Am. B*, 2010) to the already present references [24-26].

Our work takes this established technique and applies it for the first time to few-photon Rydberg EIT. We show that the complexity of the underlying Rydberg interaction transfers (with some surprises) to the propagating polaritons and that this tool greatly enhances the tunability of effective photon-photon interaction, which leads to all the results listed above. We would ask the reviewer to consider our work with this whole context in mind, and not simply reduced to a higher achievable gain of our transistor. We hope that our core messages have become more clear in our revised manuscript.

In terms of presentation of the methods and results, I find the manuscript to be lacking detail and clarity. For example, when giving measured transistor gain, neither data for extinction nor the histogram of detected clicks are shown. These should have been at least included in the supplementary information. Likewise, the description of the experimental approach and sequence are rudimentary. These deficiencies complicate accessing the quality of the measured data and make it less useful for the reader.

The aim of our manuscript is not to introduce and explain the single photon transistor in all detail, this has already been done in [18-20]. Repeating all results from these earlier works on the single-photon transistor does not help the purpose of our manuscript, namely showing how electrically-tuned Förster resonances modify and improve the Rydberg-mediated photon-photon interaction.

We do follow the advice of the reviewer and add more details on the experimental scheme in the text. Also based on comments by reviewer 2 we have added a whole paragraph describing the single Rydberg detection scheme in the *Methods* section (instead of in the supplementary methods as suggested by Reviewer #1).

Overall the manuscript contains new results and should be published. However in my opinion it does not clear the bar in terms of novelty and originality and quality of presentation for Nature Communications. I therefore do not recommend publication.

We hope that the changes we have made to the manuscript help to make clearer the different novel aspects we are presenting. We would ask the reviewer to consider all of the points listed above when reaching a conclusion about our work.

Reviewer #2 (Remarks to the Author):

This is a generally very nice presentation of an interesting experiment showing the enhancement of effective photon-photon interactions using Forster resonances.

There is one serious defect to the paper, which can be easily remedied. The paper simply cannot be understood without reading Refs 19-20. The paper heavily relies on a deep understanding of those papers.

We have addressed this problem, which was also raised by Reviewer #1. We have revised our manuscript with significantly more information both in the main text and added sections to the *Methods* to make it self-contained.

Equation (3) is asserted without even a sketch of where it comes from, or a reference for a similar calculation elsewhere. The derivation should be either described in more detail, referenced to another work that will allow it to be understood, or included as Supplemental Material. The result that the effective interaction is van der Waals in its distance dependence even in the Forster regime is interesting and important.

Our Supplementary Methods include the derivation of this equation, but the material was cited only in the beginning of the section. We now refer the reader to this information more explicitly, again in the part of equation (3). We are very happy to hear that the reviewer finds this result as surprising as we do.

With these two caveats, this is a beautiful piece of work and without question should be published in Nature Communications.

We hopefully have addressed both issues, we agree with the reviewer that this helps to make the manuscript more accessible.

Reviewer #3 (Remarks to the Author):

The authors report in this manuscript on an improved version of their single-photon transistor, ref. [19]. In particular, electrically-controlled Förster resonances are used to increase the interaction strength between gate and source photons that are stored / propagating as collective Rydberg polaritons within a cloud of Rubidium atoms. Due to the enhanced interaction, the transistor can be operated at a highly increased gain (>200), achieves high fidelities (>0.8) for detecting single Rydberg excitations, and makes it possible to investigate the coherence of the stored gate polariton after interaction with the source photons. The results are a significant improvement over the previous ones, and the work thoroughly investigates the limitations and decoherence effects of this system.

I think this work is a significant step forward towards applications of the single photon transistor as for instance in photonic circuits or for imaging single Rydberg excitations. As such, I am convinced that it will be very important and useful for other people working in this field. Thus, I believe it would be very well suited for publication in Nature Communications.

Nevertheless, I think the authors should formulate very carefully what they mean with coherence of their single-photon transistor. It seems to me that in their experiment the scattering of a single source photon completely destroys the coherence of the gate's collective Rydberg excitation (the data is below the black curve). The way I understand this data is that on average 2.7 photons are scattered, but sometimes much less than that. The remaining coherence is due to the fact that sometimes there is no photon scattered even for on average 2.7 photons.

We show in figure 3 various extreme cases, in particular the one where any incoming photon destroys the coherence completely. For this case, the retrieval efficiency would vanish even faster than we observe in the experiment. The other extreme case is that only photons scattered from the blockaded volume (which is smaller than our beam) destroy the coherence, which underestimates the decoherence. Finally, we apply the theory developed in [47], which takes into account both decoherence due to scattering as well as dephasing by polaritons which “fly by”. This approach matches very well with our data, showing that both cases contribute partially to the decoherence process.

We have clarified the paragraph discussing these limiting cases and the final result. We agree with the reviewer that our explanations were somewhat unclear and might have suggested that the coherence of the spin wave is fully depleted by a scattering event.

In my opinion the authors investigate the coherence of their single-photon transistor, but they do not show coherent operation. Nevertheless, I do understand the theory in such a way that a single scattered photon does not need to localize the position of the Rydberg excitation as long as the blockade radius is large enough (F_s increases for large r_b in Fig. 3b). Thus, coherent operation should be possible. Please note: A “coherent single-photon transistor” can also be understood as a transistor that operates coherently, i.e. a device where one uses a superposition of ground state and collective Rydberg excitation to control the transmission/reflection of the source photons. Thus, the GS would give transmitted photons, while the Rydberg excitation would result in scattered (reflected) photons, thus entangling the gate state with the path of the source photons. In any case I do not think that the expression “coherent single photon transistor” is suitable for this experiment.

We absolutely agree with the reviewer that these are two very distinct cases, which are important to separate: We investigate the coherence of the spin wave into which the gate photon is converted. We do not investigate if our transistor can be operated with a coherent superposition between two qubit states of the gate photon. Our experiments and the comparison with theory point the way to a system with preserved gate spin wave coherence during the transistor operation. We say nothing about how two qubit states would be implemented for the gate photon.

Although the terminology seems a bit muddled in the literature in general, we agree with the reviewer that any confusion should be avoided here. We thus follow the recommendation and replace the expression *coherent transistor* with the expressions *transistor with gate photon read-out* and *coherent spin-wave*.

Some further questions:

- How is the storage of the gate photon done? Is this based on EIT or just direct absorption?

We actually use both approaches, we added to the manuscript that it is based on absorption for the transistor without retrieval and light storage with EIT in the second part where the gate spin-wave is read out.

- In Figure 2(a) I think it is not good to distinguish detection fidelity and optical gain only by colour (red/blue). For people that cannot distinguish these colours very well there is no way to see what is what.

To make this plot more accessible for people who cannot distinguish between these colors, we changed the blue color to navy so the two different quantities and their corresponding axes have a different brightness, so the plot can be viewed and understood even in grayscale.

- How is the detection fidelity measured? I suppose the method uses a threshold to distinguish the states similar to Fig. 4b in ref. [19]. In my opinion this should shortly be explained in the manuscript.

We added this explanation in the *Methods* section.

- How is the accumulation of stationary Rydberg excitations observed? I guess that the authors see a temporal change of the transmission during source irradiation and then assume that this is due to the accumulation of stationary Rydberg excitations. Please formulate more logical by describing observation and conclusion.

As suggested, we rephrased this part starting with the observation: the small temporal change, followed by the conclusion: the accumulation of Rydberg excitations.

Reviewers' Comments:

Reviewer #1 (Remarks to the Author)

Reviewer #3 (Remarks to the Author)

All my previous comments are met. I can now fully recommend publication in Nature Communications.

Print Email

Resend E-mail